# High-density information storage in an absolutely defined aperiodic sequence of monodisperse copolyester

Jung Min Lee[1], Mo Beom Koo[1], Seul Woo Lee[1], Heelim Lee[1], Junho Kwon[1], Yul Hui Shim[2], So Youn Kim [2] & Kyoung Taek Kim [1]*

Synthesis of a polymer composed of a large discrete number of chemically distinct monomers in an absolutely defined aperiodic sequence remains a challenge in polymer chemistry. The synthesis has largely been limited to oligomers having a limited number of repeating units due to the difficulties associated with the step-by-step addition of individual monomers to achieve high molecular weights. Here we report the copolymers of α-hydroxy acids, poly (phenyllactic-co-lactic acid) (PcL) built via the cross-convergent method from four dyads of monomers as constituent units. Our proposed method allows scalable synthesis of sequence-defined PcL in a minimal number of coupling steps from reagents in stoichiometric amounts. Digital information can be stored in an aperiodic sequence of PcL, which can be fully retrieved as binary code by mass spectrometry sequencing. The information storage density (bit/Da) of PcL is 50% higher than DNA, and the storage capacity of PcL can also be increased by adjusting the molecular weight (~38 kDa).

[1] Department of Chemistry, Seoul National University, Seoul 08826, Korea. [2] Department of Chemical Engineering, Ulsan National Institute of Science and Technology (UNIST), Ulsan 44919, Korea. *email: ktkim72@snu.ac.kr

Unlike DNAs and proteins, the formation of synthetic long-chain molecules involves statistical uncertainties in terms of the number and sequence of monomers that join during the polymerization[1]. Synthetic polymers without statistical uncertainty in their molecular weights and sequences can store information in their chemical structures, which makes them low-cost alternatives to DNAs as molecular medium for a large-scale storage of digital information[2,3]. However, only polymers and copolymers with narrow molecular-weight distributions are produced via the living or controlled polymerization[4]. The composition of the monomers of a copolymer can be controlled, but the spatial distribution of the monomers within the copolymer is either random or completely segregated. As a result, the scalable synthesis of polymers with absolutely defined molecular weights and sequences remains a long-standing challenge in polymer chemistry[5–8].

The solid-phase synthesis has been a method to synthesize polymers with defined aperiodic sequences[9–12]. Although sequence-defined polymers with more than 100 repeating units have been synthesized[13,14], synthesis has largely been limited to sequence-defined oligomers with a limited number of repeating units due to the difficulties associated with the repeated step-by-step addition of individual monomers to achieve high molecular weights[15–26]. The iterative convergent method has been widely adopted for the synthesis of dendritic and linear macromolecules with precisely defined chemical structures[27–30]. In this process, the coupling product becomes a constituent unit for the next iteration of the coupling reaction. Therefore, the number of repeating units grows exponentially as does the molecular weight of the resulting polymer without distribution[31–37]. Although high molecular-weight polymers without molecular-weight distribution can be synthesized efficiently by the convergent method, the self-iterative nature of the coupling step hinders the convergent synthesis of monodisperse oligomers and polymers with perfectly defined aperiodic sequences composed of two or more chemically distinct monomers. For this reason, only oligomers and block co-oligomers with repetitive or palindromic sequences have been synthesized by the convergent pathway[37–39].

Here we report the synthesis of poly(α-hydroxy acid) (PAH) composed of a large discrete number of monomers in an absolutely defined aperiodic sequence. We demonstrate that digital information can be stored in the aperiodic sequence of poly(phenyllactic-co-lactic acid) (PcL), built via the direct translation of binary code to the chemical structure built with four dyads of monomers as constituent units. This process was named to the 'cross-convergent' method (Fig. 1). The stored 64-bit word (SEQUENCE) can be fully retrieved as binary code in a single tandem mass spectrometry sequencing. To expand the ability to read the stored information, we also devised a sequencing method for PcLs with a large number of repeating units (128 units) by measuring the mass of fragments produced by the hydrolysis of the polymer backbone. PcL can be synthesized in a minimal number of coupling steps from reagents in stoichiometric amounts, and its digital information storage density (bit/Da) is 50% higher than that of DNA[40–43]. The molecular weights and production quantities of the reported PAHs are scalable, and no statistical uncertainty is associated with either molecular weight or sequence. The reported PcL could serve as a molecular medium for the storage of digital information. In addition, sequence-defined monodisperse polymers could contribute to the exploration of new properties and functions of polymers arising from the unlimited diversity of their chemical structures.

## Results

**Synthesis of PcL.** To solve the inability of the iterative convergent method to address an aperiodic sequence of a polymer, we devised a cross-convergent pathway to define an aperiodic sequence of monomers comprised of a copolymer (Fig. 1). The chemical sequence of a copolymer consisting of two monomers can be converted directly to a digital binary code. We used a binary code indicating a word 'SEQUENCE' to construct a copolymer with an absolutely defined aperiodic sequence of two monomer. In this work, we used rac-phenyllactic acid (P) and rac-lactic acid (L) as the monomers representing 0 and 1, respectively. Any sequence of a copolyester, poly(phenyllactic-co-lactic acid) (PcL), can be expressed as a combination of the permutations of its constituent monomers. For a binary sequence, the permutations of 0 and 1 give four combinations (00, 01, 10, and 11), which can be translated to the chemical structures composed of two monomers (dyads).

Four dyads encompassing all possible permutations of the binary sequences (PP, PL, LP, and LL) were synthesized by coupling α-hydroxy acids orthogonally protected by benzyl ester and t-butyldimethylsilyl (TBDMS) ether[33–35,44]. The dyad PP was quantitatively converted to HO-PP-Bz by the selective removal of the TBDMS group with boron trifluoride etherate (BF$_3$·Et$_2$O) at room temperature. Hydrogenation of the benzyl ester in the dyad PL with Pd/C generated TBDMS-PL-COOH in high yield (>95%). The equimolar mixture of TBDMS-PL-COOH and HO-PP-Bz was subsequently coupled to form a tetrad, PLPP by esterification. These orthogonal deprotection and coupling steps constituted one iteration of the convergent growth of PcL (indicated by the green box in Fig. 1). The same procedure was repeated to synthesize all five tetrads by cross-converging the required dyads. The tetrads were sequentially converged to form 8-bit characters (S, E, Q, U, N, C), and joined to form the full 64-bit word consisting of eight ASCII characters (SEQUENCE).

The chromatographic separability of PcL and its constituent units on silica stationary phases diminishes as the molecular weights of the compounds increase[45]. This renders a mixture of constituent units and the desired high-molecular-weight PcL inseparable (Supplementary Fig. 1). However, the difference between the molecular weight of a PcL and its constituent units persists through all iterations of convergent coupling due to the exponential growth of the molecular weight of the coupled product (Fig. 2a, b). Therefore, the monodisperse PcLs with greater than 16 repeating units were purified by preparative size-exclusion chromatography (prep-SEC) in a recycling mode with a loading of up to 1 g per separation. This purification method allowed us to obtain monodisperse PcLs in high yield with stoichiometric amounts of reagents in a scalable manner with regard to both molecular weight (>38000 Da) and quantity (>1 g). This purification method can also be applied to any set of monodisperse polymers with a difference in a hydrodynamic volume, for example, linear and cyclic polymers having the same chemical composition and molecular weight as the intramolecular cyclization of a polymer chain reduces the hydrodynamic volume, which translates to a lower molecular weight on SEC[46].

Taking the redundancy of the tetrads and the frequency of the letter E into account, 18 convergent coupling steps with stoichiometric amounts of reagents were required to synthesize sequence-defined PcL from four dyad constituent units. The successful synthesis of monodisperse PcL storing a 64-bit binary code was confirmed by $^1$H and $^{13}$C NMR, GPC and MALDI-TOF MS (Fig. 2 and Supplementary Figs. 2–6). Due to the exponential growth of PcL via the cross-convergent pathway, no deletion errors, accumulation of polymer chains missing terminal residues, were detected by MALDI-TOF MS analysis of PcL composed of 64 or 128 repeating units (Fig. 2c). Data from each analysis, in particular MALDI-TOF MS, indicated that the synthesized PcLs were monodisperse in molecular weight and free of lower-molecular-weight impurities.

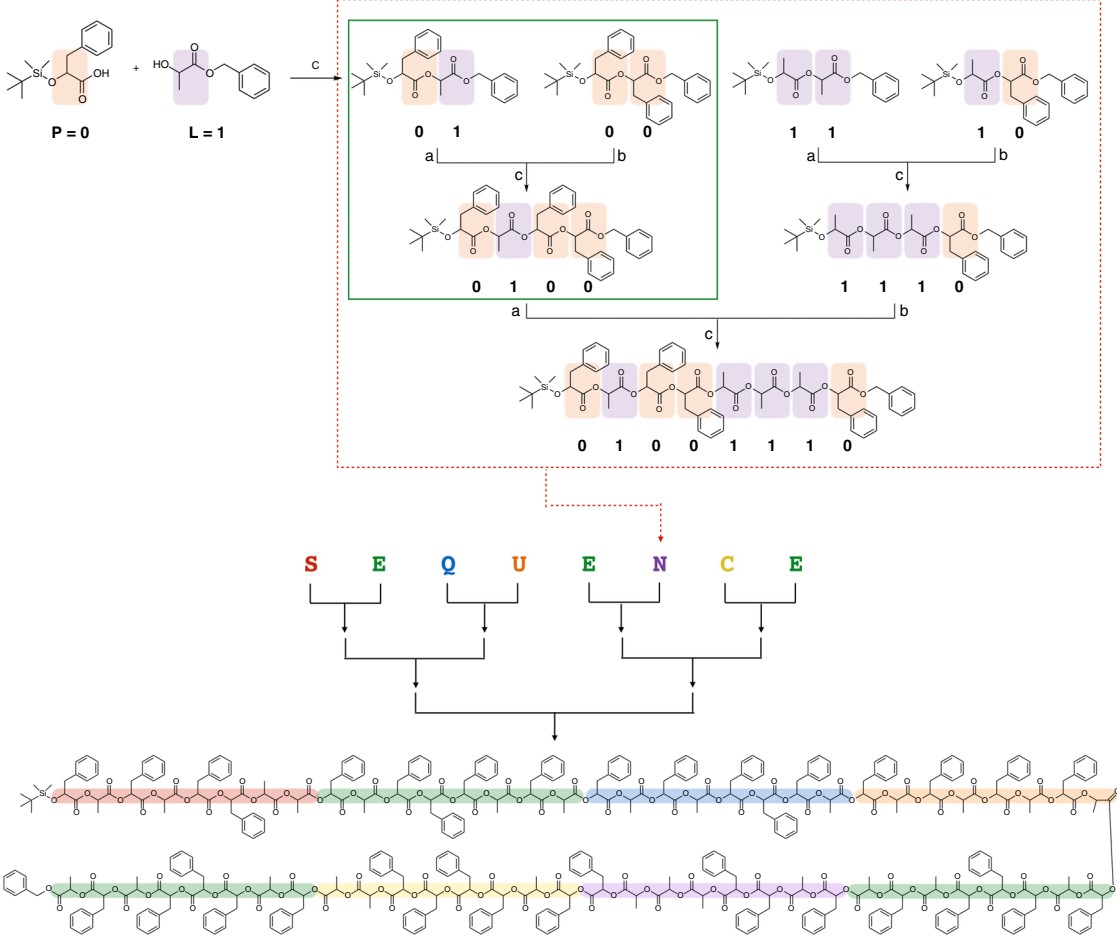

**Fig. 1 Writing information in sequence-defined PcL.** Schematic illustration of the cross-convergent strategy to synthesize PcL with a sequence corresponding to the binary code for the 64-bit word SEQUENCE. Four dyads of phenyllactic acid (P representing 0) and lactic acid (L representing 1) were used as the constituent units of PcL. The cross-convergent pathway with four dyads could express any aperiodic P and L sequence. Tetrads were sequentially converged to form one-byte characters (shown in a red box), two-and 4-byte words, and finally PcL containing the 64-bit digital information encoding SEQUENCE. The green box indicates a set of deprotections and coupling constituting a convergent growth (**a** BF$_3$·Et$_2$O, RT, 6 h; **b** Pd/C, H$_2$, RT, 8 h; **c** EDC·HCl, DMAP, 8 h).

**MALDI-TOF mass sequencing of PcL.** The PcL sequence was decoded by a tandem mass technique using MALDI-TOF MS/MS in a positive-ion mode because of its high signal-to-noise ratio and its ability to detect the fragmentation patterns of high-molecular-weight PcLs (<10 kDa). Fragmentation occurred at the C($\alpha$)-O bond of the PcL backbone, which produced a$i$ fragments that contain the original alpha group (TBDMS) and a new carboxylic acid terminus and y$i$ fragments that contain the original omega group (benzyl ester) and a new alkene terminus[47]. Each fragment series had mass difference by 72 Da (residue L) or 148 Da (residue P) in decreasing order relative to the peak corresponding to [M + Na]$^+$ ion (Fig. 3a, b). A single mass spectrum was sufficient for decoding all the information in the 64-bit PcL (Fig. 3c). This was because a single mass spectrum showed a series of fragments that can be read in opposite directions, one from the TBDMS-terminus to the benzyl-terminus, and the other from the benzyl-terminus to the TBDMS-terminus. This also enhanced precision, because two retrieved sequences containing the same information could be compared (Supplementary Figs. 7–32 and Supplementary Tables 1–13). The chemical sequence of the PcL was directly converted to binary code designating the word SEQUENCE as highlighted in the red box in Fig. 3.

PcLs with high molecular weights (>64 repeating units) could not be directly sequenced under our tandem mass condition with MALDI-TOF MS/MS presumably due to the inability to analyze high molecular-weight parent ions by the instrument. To overcome this limitation, we devised a degradative sequencing method using MALDI-TOF mass spectrometry because of higher molecular-weight limit (~20 kDa) for the desorption of polymers. Poly($\alpha$-hydroxy acid) (PAH) could easily be degraded to their constituting monomers via hydrolysis of ester groups in the polymer backbone. The random hydrolysis of ester groups along the polymer backbone would create the fragments of the parent PcL, which could be directly detected by MALDI-TOF mass spectrometry. Therefore, the sequence of monomers in PcL could be decoded by reading the mass difference between adjacent fragments from the mass spectrum.

Utilizing facile degradability of PcL[48–50], we chemically degraded a 128-bit PcL containing a 16-letter word (SEQUEN-CESEQUENCE) via the hydrolysis of the ester groups in the main chain. In the presence of NaOH (0.3 equivalent to the ester groups), 128-bit PcL was incubated in THF for 30 min at 70 °C. MALDI-TOF mass spectra of the hydrolyzed PcL showed a series of mass peaks in decreasing order from the mass of the molecular

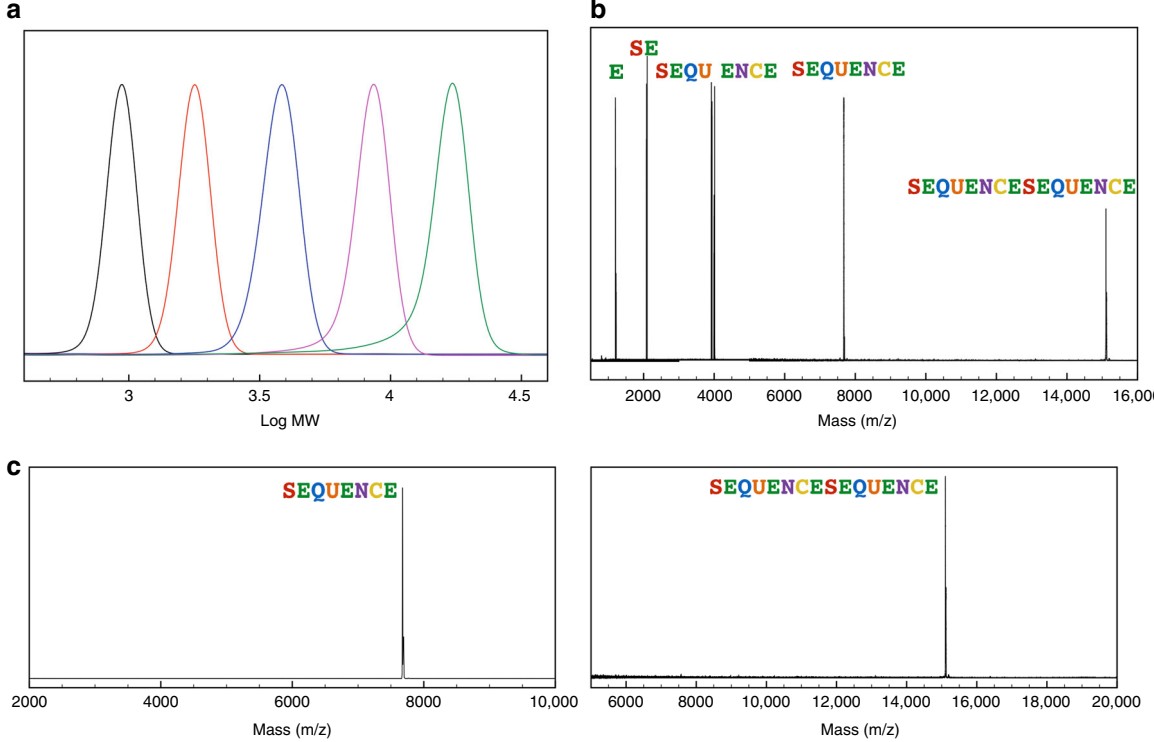

**Fig. 2 Molecular-weight analysis of PcL. a** Gel-permeation chromatography (GPC) analysis of PcLs with 8, 16, 32, 64, 128 repeating units. **b** Combined MALDI-TOF mass spectra of PcLs encoding the letter (E, 1201.5 Da), two-letter word (SE, 2082.9 Da), four-letter words (SEQU, 3920.4 Da and ENCE, 3996.5 Da), 64-bit word (SEQUENCE, 7674.5 Da), and 128-bit word (SEQUENCESEQUENCE, 15105.3 Da). The peak was assigned to a $[M + Na]^+$ ion. **c** MALDI-TOF spectra of 64- and 128-bit PcL showing no deletion errors or contamination of lower molecular-weight fragments.

ion (Fig. 4a and Supplementary Figs. 33, 34). The mass spectrum could be decoded to binary code by reading the mass difference between the peaks in a direction from the TBDMS-terminus to the benzyl terminus. The full sequence except ultimate 8 residues (residue 1–120) could be deciphered utilizing a series of MALDI-TOF mass spectra of the degraded PcL covering the entire molecular-weight range (1000–16,000 Da) (Supplementary Table 14). The last eight residues (residue 121–128) close to the Bz-terminus of the PcL were subsequently sequenced by a tandem mass spectrum of one of the lower molecular-weight fragments (x22, $m/z$ of a parent ion = 2703 Da in MALDI-TOF, Supplementary Fig. 35) in order to remove the noise from the matrix molecules in MALDI-TOF mass (Supplementary Table 15). The decoded sequence of the 128-bit PcL coincided with the chemical structure of the PcL without any error (Fig. 4b).

The density of information storage, defined as the number of bits per unit mass, was 0.009 bit/Da in PcL. This was 50% higher than the storage density of DNA (0.006 bit/Da)[40–44]. Due to the simplicity in chemical structures, our results suggest that the cost of synthesizing sequence-specific PcLs could be substantially lower than the cost of writing information on DNAs. Thus, PcLs could provide an alternative to DNA and molecular media for archival storage of large amounts of information.

**Scalability of the convergent synthesis of PAH.** The storage of large information in molecular media demands the original bit-information to be divided to a unit size (for example, 64 or 128 bit), so that the divided information can be stored in multiple polymer chains. Each polymer chain is required to store the unit data along with the address information, so that the original information is fully restored by the integration of decoded sequences of all polymer chains. Therefore, one of the

requirements for synthetic polymers as information storage media is a synthetic method for polymer chains having a number of repeating units that is sufficiently large to meet the required capacity of information storage.

To demonstrate the scalability of the convergent method in terms of molecular weight, we synthesized monodisperse poly(*rac*-phenyllactic acid) (PA*n*) composed of a large number of repeating units (Fig. 5a). Here, *n* represents the number of repeating units. Starting with tetramers PA4 (10 g, 12.27 mmol), seven iterations of convergent growth yielded PA256 with molecular weight of 38175 Da (experimentally found at 38191 Da) in an overall yield of 15% (1.07 g, 0.028 mmol) (Fig. 5c, d). PA512 with a molecular weight greater than 70 kDa could only be synthesized in low yields (<5%) even after an extended period of time for the coupling reaction. This could be a result from the difficulties of finding the chain end of high molecular weight PA*n* under the conditions we employed for the esterification. The convergent growth strategy also allowed us to synthesize monodisperse PAHs with any number of repeating units by changing the constituent units in the final coupling step. For example, PA80 and PA96 were synthesized by selecting the constituents that afforded the desired number of repeating units (Fig. 5b). Our results indicated that the number of repeating units of PAH could be optimized to store information with a desired length and format suitable for a large-scale storage of digital information.

**Discussion**
In summary, we synthesized monodisperse poly(α-hydroxy acid)s (PAHs) composed of two chemically distinct monomers in absolutely defined aperiodic sequences via the cross-convergent method. Our proposed method allowed to synthesize polymers

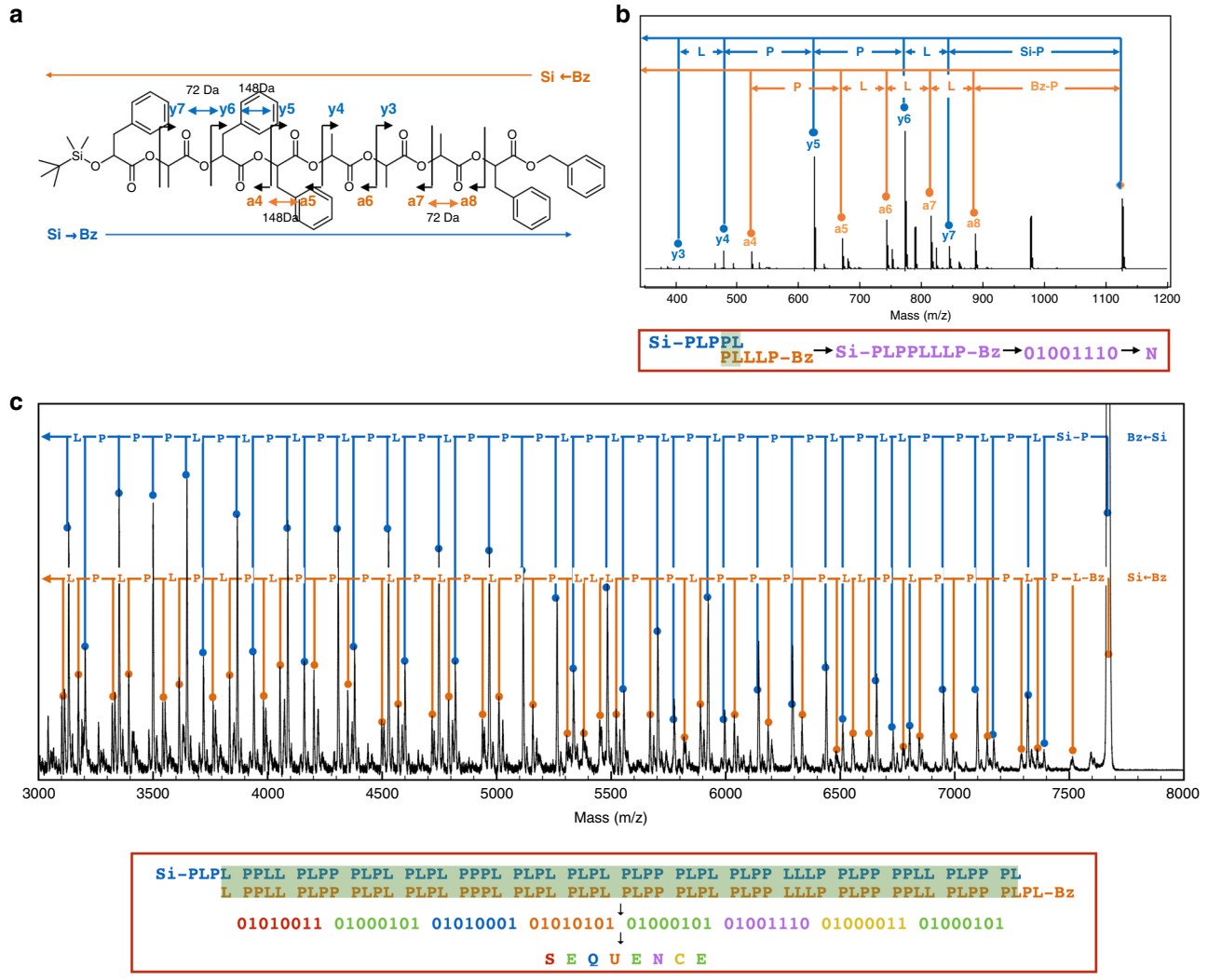

**Fig. 3 Decoding of sequence-defined PcL by tamdem mass spectrometry. a** Fragmentation of 8-bit PcL under MALDI-TOF MS/MS experiments showing a series of $a_i$ and $y_i$ fragments. **b** MALDI-TOF MS/MS spectrum of a PcL, in which 8-bit information corresponding to the letter N was stored. Two series of fragments ($a_i$ and $y_i$ fragments) could be read simultaneously in the spectrum. The PcL sequence was decoded by reading the spectrum in both directions relative to the molecular ion peak [M + Na]$^+$. The deciphered chemical sequence (Si-PLPPLLLP-Bz) was converted to digital code (01001110), which represented the letter N. **c** Tandem mass sequencing of the entire 64-bit information stored in the PcL. The entire chemical sequence was decoded using a single mass spectrum, followed by conversion to digital information to read the word SEQUENCE (red box).

with any aperiodic sequence by utilizing the permutation of two monomers (dyads) as constituent units for the convergent synthetic pathway. The synthesis of monodisperse and sequence-defined PAHs with large numbers of repeating units was accomplished with notably fewer synthetic steps than were required for solid-phase synthesis of the same polymers. In addition, only stoichiometric amounts of constituent units were needed to synthesize a monodisperse product. This is in contrast to solid-phase synthesis, which requires a large excess of reagents to prevent incomplete coupling during the introduction of individual monomers. We demonstrated that digital information can be stored in a sequence of poly(phenyllactic-co-lactic acid) (PcL), which can be fully retrieved by a single measurement of MALDI-TOF tandem mass spectrometry. Utilizing facile degradability of ester groups in the PcL, the MALDI-TOF mass sequencing of high molecular-weight PcL allowed a sequential read of the stored information. Information storage density of PcL (bit/Da) is 50% higher than that of DNA, which render the PcL to be an alternative molecular media for storing digital information.

Given the wide availability of α-hydroxy acids with different substituents and stereochemical configurations, the monodisperse and sequence-defined PAHs reported here should provide exciting opportunities to explore an unlimited diversity of chemical structures and the consequent properties and functions of synthetic polymers[51–53]. Our results also suggest that large-scale preparation of monodisperse polymers with precisely defined sequences and desired molecular weights is possible if executed in an automated and parallel fashion[42].

## Methods

Synthetic procedures and characterization of PcLs and PAHs used in this work are described in Supplementary Information.

**SEC of monodisperse PAH**. Size-exclusion chromatography (SEC) of PAHs was conducted by injecting 5 mL of a PAH solution in CHCl₃ (100 mg mL⁻¹) to a Recycling Preparative HPLC (LC-9260 NEXT, Japan Analytical Industry) system equipped with JAIGEL-2.5 H/2 H/3 H columns and a differential refractometer. Chloroform was used as an eluent with a flow rate of 3.5 mL min⁻¹. Before injection, the solution was filtered through a PTFE syringe filter (Whatman, 0.2 μm

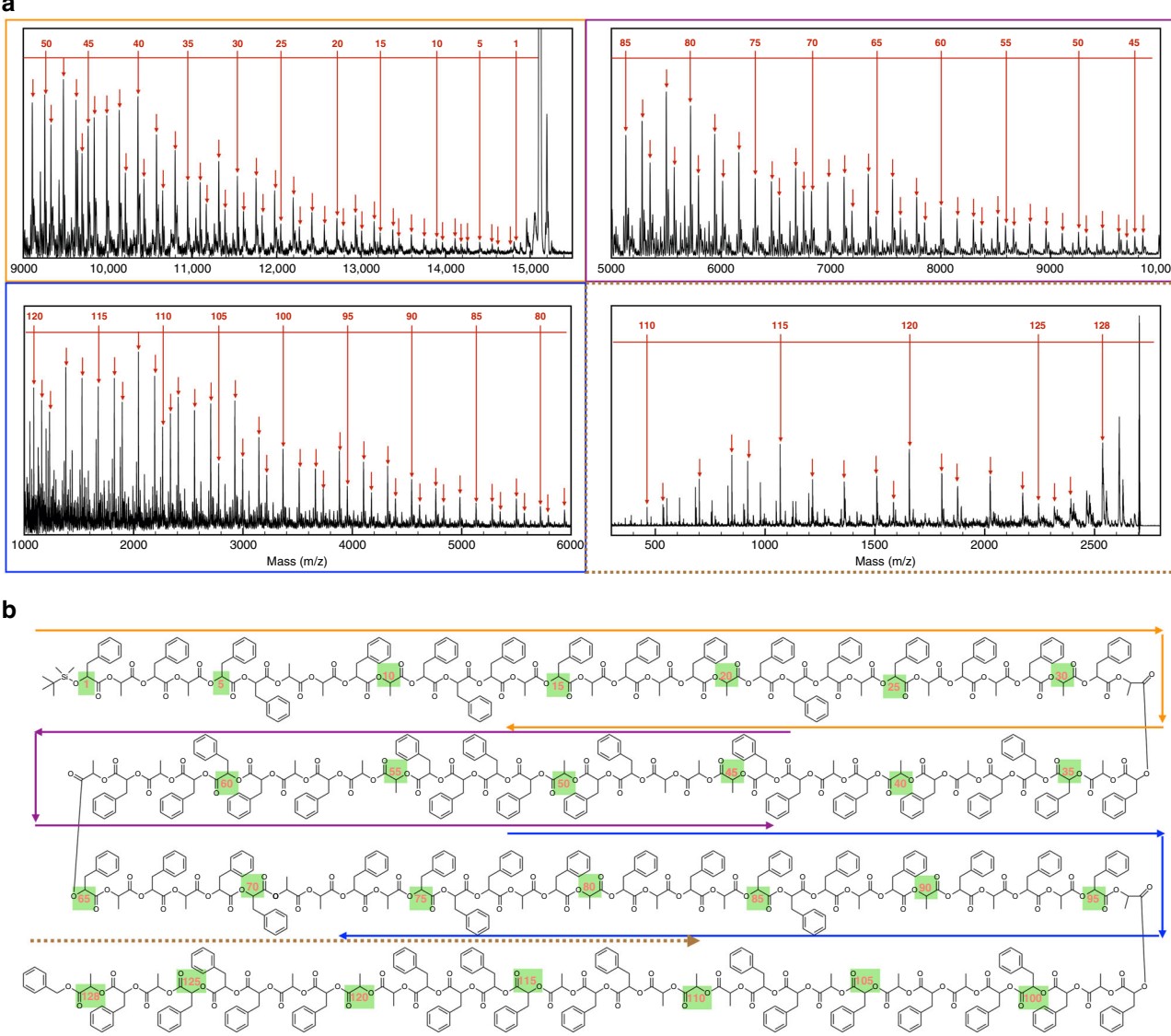

**Fig. 4 MALDI-TOF mass sequencing of 128-bit PcL. a** A series of MALDI-TOF mass spectra of chemically degraded PcL via hydrolysis. The assigned peaks are marked by arrows. The mass spectra of $xi$ fragments were used for MALDI-TOF sequencing. The sequence of the last 8 repeating units at the Bz terminus was decoded by MALDI-TOF MS/MS to avoid the noise from the signal of the matrix molecules. **b** The chemical structure of 128-bit PcL drawn with the decoded sequence. The repeating units are numbered in an increasing order from the first repeating unit (P) at the TBDMS terminus. The decoded sequence is identical to the encoded chemical structure.

pore). The SEC was performed under a cycling mode until the coinciding peaks were separated. The desired fraction was collected using a fraction collector. Two prep-SEC systems ran in parallel, giving the maximum capacity of separation of 1 g.

**MALDI-TOF and Tandem mass sequencing of PcL**. Molecular weights of PcLs and their fragments were measured on a Bruker Ultraflex TOF/TOF mass spectrometer equipped with a smartbeam 2 (Nd:YAG laser) at 2000 Hz (MALDI-MS) or 1000 Hz (MALDI-MS/MS). For MALDI-MS analysis, the instrument was operated in a positive reflector mode with voltage for ion source 1 (20 kV), ion source 2 (17.65 kV), lens (8.4 kV), reflector 1 (21.2 kV) and reflector 2 (10.65 kV). Voltage for ion source 2, lens, and reflector 2 is raised up to 17.75 kV, 8.8 kV and 10.8 kV depending on the molecular weight of a polymer. PAHs with molecular weight above 20 kDa were analyzed in a positive linear mode with the voltage for ion source 1 (20 kV), ion source 2 (18.8 kV), and lens (8.0 kV). External calibration was based on peptide and protein (ProteoMass Peptide/Protein MADLI-MS Calibration Kit (mass to charge ratio from 750 to 66000 Da), Sigma). Tandem mass sequencing (MS/MS) was performed in positive reflector mode with acceleration voltages for ion source 1 (7.62 kV), ion source 2 (6.8 kV), lens (3.6 kV), reflector 1 (29.5 kV), reflector 2 (13.9 kV), LIFT 1 (19.00 kV), and LIFT 2 (2.85 kV) using no gas option. The precursor ion was used as internal calibration. For MALDI and MS/MS analysis, 2-(4-

Hydroxyphenylazo)benzoic acid (HABA) or trans-2-[3-(4-tert-Butylphenyl)-2-methyl-2-propenylidene]malononitrile (DCTB) was used as a matrix. A polymer sample and matrix were dissolved in THF at 5 mg mL$^{-1}$ and 30 mg mL$^{-1}$, respectively, and, these solutions were mixed in 1:1 to 1:5 ratio depending on the molecular weight of the analyte. 0.8 µL of the mixed solution was spotted on a MALDI plate, and dried in the air. PcLs were fragmented into a$i$ fragment containing the original alpha group (TBDMS) and a new carboxylic acid terminus and y$i$ fragment having the original omega group (benzyl ester) and a new alkene terminus via 1,5-H rearrangement. For sequencing, the mass difference between adjacent fragments (72 Da for L residue and 148 Da for P residue) was used.

**Chemical degradation of PcL**. A vial was charged with 128-bit PcL (25 mg, 1.6 µmol) and THF (4 mL). To this solution, 0.5 M NaOH solution (150 µL, 47 eq. to the PcL) was added. The vial was tightly sealed and heated to 70 °C. A portion (0.6 mL) of the solution was taken at a 30 min interval, which was subsequently diluted with ethyl acetate (10 mL). The diluted solution was washed with brine (5 mL). The combined organic layer was dried with MgSO$_4$ and concentrated in vacuo. The chemical degradation of PcL was confirmed by GPC using DMF as an eluent. Upon hydrolysis, PcL is dissociated into two fragments. One is the c$i$ fragment that contains the original alpha group and a new carboxylic acid

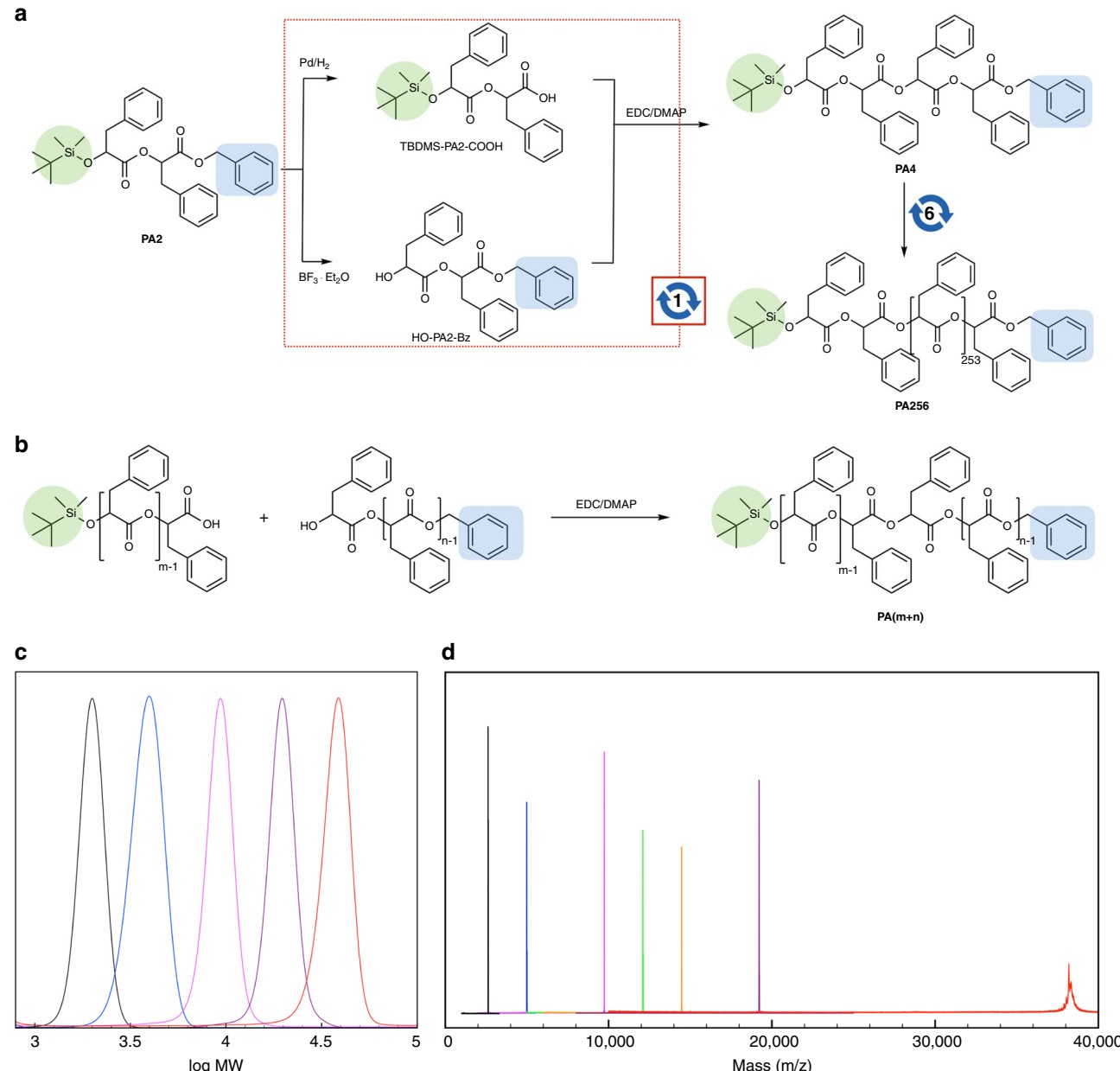

**Fig. 5 Molecular-weight analysis of monodisperse PAHs. a** The iterative convergent synthesis of poly(*rac*-phenyllactic acid) (PA*n*). The deprotection and subsequent esterification reactions shown in the red dotted box constitute a convergent growth step. The number shown with circular arrows represent an iteration of the convergent growth step. **b** Method to obtain the monodisperse PAHs with desired number of repeating units via combination of constituent units. TBDMS and Bz protecting groups are highlighted in green and blue, respectively. **c** Gel-permeation chromatography of PAs with 16, 32, 64, 128, and 256 repeating units. **d** Combined MALDI-TOF mass spectra of PA16 (black, 2615.2 Da), PA32 (blue, 4986.4 Da), PA64 (magenta, 9728.4 Da), PA80 (green, 12100.7 Da), PA96 (orange, 14474.0 Da), PA128 (purple, 19220.6 Da), and PA256 (red, 38191 Da). MALDI-TOF MS of PA256 was measured in a linear mode. Other samples were measured in a reflection mode. The peak was assigned to a $[M + Na]^+$ ion.

terminus. The other is the $xi$ fragment that contains the original omega group and a new hydroxyl terminus. The mass peaks of $ci$ fragments were found to exhibit low intensities in MALDI-TOF. Therefore, the peaks corresponding to $xi$ fragments were used for sequencing. For last eight residues, a fragment of PcL (mass to charge ratio, 2703.9 Da) was used as a parent ion for the analysis by tandem mass.

## Data availability

All data are available from the authors upon request.

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

## Acknowledgements

This work was supported by National Research Foundation (NRF) of Korea (NRF-2019R1A2C3007541). S.K. acknowledges the support by NRF of Korea (NRF-2018R1A2B6008319). K.T.K. thanks Seoul National University (SNU) for the support by Creative-Pioneering Researchers Program (305-20190050).

## Author contributions

K.T.K. conceived and supervised the project. J.M.L., M.B.K., S.W.L., H.L., and J.K. synthesized monomers and polymers. J.M.L. performed and analyzed tandem mass spectrometry. Y.H.S. and S.Y.K. conducted rheological measurements and analyzed data. All authors contributed in drafting the manuscript.

## Competing interests

The authors declare no competing interests.
