## [Peer Review File · Nature Communications]

Reviewers' comments:

Reviewer #1 (Remarks to the Author):

This manuscript reports a relevant synthesis approach to large sequence-defined polymers, and would nicely contribute to the efforts currently devoted to this new area of research. However, prior to being considered for publication in Nature Communications, it requires extensive revision in order to address the major concerns listed below.

In the proposed synthesis method, four dyad-containing monomers were first prepared by using different combinations of phenyllactic acid (P, defined as the 0-bit) and lactic acid (L defined as the 1-bit), hence yielding 00, 01, 10 and 11 coded dyads. Using a cross-convergent method, these dyads were then employed to prepare different reagents containing 4 bits of information i.e., tetrads), further reacted together to lead to byte-containing macro-monomers (i.e., reagents composed of 8 co-monomers) that were ultimately used to prepare long so-called PcL chains. As a result, these chains have a DP which is a multiple of 8, well suited to use the ASCII code to write information. Instead of using a (very) large library of reagents, this approach relies on the initial synthesis of the appropriate series of byte-containing macro-monomers needed for the preparation of any targeted final sequence, which indeed minimized the number of coupling steps to prepare long sequence-defined polymers.

At this point, the authors should better define the size limit they used to distinguish (co)oligomers from (co)polymers. In the abstract (line 3) as well as in the Introduction (page 2, second paragraph), they wrote that synthesis of sequence-defined species has been so far limited to oligomers and claimed that their method enabled preparation of polymers, the first example being chains of DP 64. However, sequenced-defined chains in this size range (as well as above) have already been reported a few years ago (e.g., works from the Lutz group reported in reference 18 as well as in Nat. Commun. 2017, 8, 967 which was omitted).

Another clear advantage of this cross-convergent approach was that any unreacted macro-reagent can readily be separated from targeted polymers since the mass of the latter ones increased exponentially. As a result, raw samples could easily be purified to yield highly monodisperse samples. This should be better acknowledged by the authors. First, the verb "synthesize" should be replaced by "obtain" in the sentence (page 5) "This purification method allowed us to synthesize monodisperse PcL in high yield ...". Second, caption of Fig. 2b-d should be clarified: were data in Fig. 2b obtained for a raw sample prior purification and mass data shown in Figs. 2c-d recorded after purification of this sample?

Mass spectrometry was employed here as a characterization technique, using MALDI to generate sodium adducts of PcL in the gas phase. Surprisingly, according to the experimental section, no sodium was supplemented to the MALDI samples: this could have helped promoting better ionization of the largest chains and hence improve their MS/MS sequence coverage. Moreover, low measurement accuracy achieved when using MALDI-MS(/MS) does not permit to report m/z values with the precision provided in Supplementary Tables 1-15: m/z values may be reported with one decimal digit (certainly not two). Moreover, it is not clear whether unit resolution was achieved for all ions and whether ions were measured at their monoisotopic values. For example, in Tables 7-9, while the $[M+Na]^+$ species is expected at a monoisotopic m/z value of 2081.8, its maximum isotopic value is m/z 2082.8: according to the reported experimental m/z of 2082.3, one can either suspect that resolution was too low to distinguish isotopes or that there was an issue with mass calibration. Similar issues are encountered in Table 10 (isotopic maximum calculated at m/z 2158.8). In contrast, calculated values reported for $[M+Na]^+$ in Tables 11 and 12 are wrong. In Table 11, this should be m/z 3018.4 (monoisotopic) or m/z 3920.4 (at isotopic maximum). In Table 12, this should be m/z 3994.4 (monoisotopic) or m/z 3996.4 (at isotopic maximum). With this regard, data of all Supplementary Tables have to be thoroughly revised. Similar remarks apply for m/z values reported for species shown in Figure 5, particularly those measured in the TOF linear mode that exhibit very large error. Moreover, data in the caption of this figure are very misleading as it appeared that m/z values were sometimes mixed with mass values.

Because the two coded bits had different masses (P: 148 Da, L: 72 Da), MS/MS could be employed as a sequencing method. A nomenclature is available to designate MS/MS fragments of synthetic polymers (see Wesdemiotis et al *Mass Spectrom. Rev.* 2011, 30, 523) and should be employed here. According to this nomenclature and owing to the bond cleavage experienced by P_cL, fragments containing the left-hand side (alpha) termination should be named *a_i* while fragments containing the right-hand side (omega) termination should be named *y_j* (with *i* and *j* the number of monomers they contain, respectively). Accordingly, analysis of *a_i* fragments should allow the sequence chain to be partially re-constructed from the left to the right, whereas *y_j* permitted to partially decipher 0/1 bits from the right to the left. This partial sequence coverage is not really acknowledged in the dissociating scheme shown in Figure 3a. It also means that both fragment series need to be considered to recover the whole polymer sequence, even for quite small species containing one byte of information. Details reported in the experimental section are not sufficient to figure out how MALDI ions were activated and whether the activation energy was increased with the polymer DP. In addition, data reported in Supplementary Tables are confusing in terms of sequencing. For the sake of simplicity, let's take the example of Supplementary Table 1 reporting MS/MS data recorded for the 01010011 chain (coding for S). First, cleavage of the C-O bond should proceed according to a rearrangement which leads to the transfer of one proton from the right- to the left-hand of the dissociation bond, resulting in i) *a_i* fragments that contain the original alpha group and a new OH termination and b) *y_j* fragments that contain the original omega group and a new termination which depends on the nature of the unit before which cleavage occurred: –CH=CH₂ if cleavage occurred before L or –CH=CH₂-Ph if cleavage occurred before P. Detection of one or the other fragment series depends on which part of the cleaved chain is the adducted sodium. This is not “bidirectional fragmentation” but dissociation that leads to complementary fragments. On the one hand, fragments annotated in orange at *m/z* 523, 595, 743, 891 and 963 respectively correspond to *a₄*-*a₈* fragments (calculated *m/z* values are false) and *m/z* difference measured between these ions allowed the five last units to be identified as L, P, P, L and L. Again, these fragments contain the original alpha end-group and allow this partial sequence to be re-constructed from the left- to the right-hand side. On the other hand, fragments annotated in blue at *m/z* 405, 553, 625, 773 and 845 respectively correspond to *y₃*-*y₇* fragments (calculated *m/z* values are false) and *m/z* difference measured between these ions allowed the five first units of the original chain to be identified as P, L, P, L and P (the first P unit being identified from the mass difference between [M+Na]⁺ and *y₇*). In summary, in great contrast to annotations found in Supplementary Table 10, blue ions contain the original omega termination and allow the sequence to be reconstructed from the right to the left. This aspect has also to be corrected in all Supplementary Tables as well as in explanations provided for sequencing in the main text.

However, this fragmentation pattern did not allow reliable sequencing of the longest chains since there is no part of the sequence covered by both *a_i* and *y_j* fragments. The authors wrote that the upper DP for reliable sequencing was 90 but reported data provided evidence for sequencing of chains only up to DP 64. As a result, the polymer with DP 128 had to be chemically degraded and the implemented hydrolysis reaction was shown to proceed from the alpha to the omega termination, yielding a series of products that differ from one another by the mass of a single unit that could be identified in MS. Moreover, MS/MS still had to be performed to identify the last byte of information. Although very interesting from the analytical point of view, this directional hydrolysis pathway was neither explained nor rationalized. Still, the need for performing such a chemical degradation to read messages of more than 64 bits remains a main drawback for these P_cL polymers. In other words, this manuscript does report easy sequenceability for polymers of larger size than other species reported in the literature. In this context, the last sentence of the abstract which deals with a perspective not demonstrated in the present manuscript should hence be removed; instead, the authors should acknowledge that full MS/MS sequence coverage could only be demonstrated up to DP 64 and that a sample pre-treatment had to be performed for partially selective cleavage of longer chains prior subjecting so-obtained products to MS and MS/MS analysis.

The last part of the manuscript deals with the synthesis of even larger polymers but appears somehow out of scope of the study since this is no longer about sequence-defined species but homopolymers made of P or L units and study of some of their macroscopic properties.

Reviewer #2 (Remarks to the Author):

The manuscript describes a convergent approach to sequence-encoded macromolecules via esterification reactions. Monodisperse macromolecules with up to 512 repeat units are synthesized, and a molecule with the binary code for the word "sequence" is presented. While the present approach by itself might be novel, and surely a lot of work was dedicated to reach this aim, I am a bit afraid that the approach is not overly novel. The binary system in use has already been introduced by Lutz and coworkers several years ago, with similar coding lengths being described. The present chemistry requires protecting group chemistry (even if it is a simple one), while other groups (e.g. Du Prez or Meier) have shown that orthogonal protecting-group free approaches are largely superior. Also the idea to synthesize smaller fragments that are then coupled later has been shown (qr codes by Du Prez) before.

Characterisation of the sequences, as much as the determination of physical parameters of the discrete materials follows literature examples and delivers nothing beyond the already known (the authors might also want to check De Neve, *Angewandte Chemie*, 2019 for further literature). In other words, I think this is good work, and I have no real detailed comments at this stage (which doesn't mean there wouldn't be any), but don't think that this manuscript is really novel. Neither the approach is new, nor the chemical system or the analysis of the polymers is.

Reviewer #3 (Remarks to the Author):

Kim and coworkers show the synthesis of discrete and sequence defined phenylactic-co-lactic acid polymers using a cross-convergent method resulting in a nicely scalable method. Overall, this is a nice paper where they show how these discrete polymers with an aperiodic sequence can be used as for data storage, an ever growing field of interest. The results are promising as these polymers are relatively cheap (compared to DNA), even though the synthesis and purification can be time consuming. Hence, this work contributes towards solving the data storage problem in the future and is therefore of importance to a broad audience. The paper is recommended for publication after major revision, where the following points are addressed in a convincing way.

A structure-property relationship for the high Mw lactic acid polymers is shown. This is interesting, though, it would be more interesting to show this for the copolymers. The manuscript is focused on the copolymers and therefore it is much more important to show the structure-property relationship of the copolymers. Then, the question on the importance of the sequence is shown not only for data storage but also for polymer properties. Showing only the rheological measurements of the LA polymers does also not fit in the flow of the manuscript. One could then answer the question of on what the effect is of the exact sequence on the polymer properties. Would you be able to also perform the rheological measurements on the copolymers and elaborate on the importance of the exact sequence for polymer properties?

In the second paragraph of the introduction, the authors describe ways to make discrete macromolecules or oligomers/polymers. They describe (cite) many examples of linear macromolecules synthesized via an iterative convergent method. However, the authors use the cross-convergent method and it would be beneficial to know a bit more about this method before explaining the synthesis.

The authors show that the synthesis of the polyester can be nicely performed in an iterative way. However, can the authors elaborate on the choice for racemic lactic acid and phenylactic acid?

Would similar results be obtained regarding data storage when only one of the two monomers was chosen, yet, controlling the stereochemistry?

What is the reason why PA could be synthesized up to 256 repeating units while the LA could be synthesized up to 512 repeating units in an efficient way?

The rheological measurements of the LA polymers with 128 or more repeating units indicate the presence of entanglements in the bulk. Does this imply that all polymers with a lower molecular weight do not have any entanglements? Where is the point for having entanglements since the M_e was measured to be 3920 Da?

List of corrections and additions

1. New references were added to the reference section.
2. All changes and corrections in the manuscript are highlighted.
3. All mass values were remeasured for the revision. The manuscript and supplementary information were fully revised with these data.
4. All sequencing experiments were reperformed to enhance the accuracy with recalibrated MS/MS and MALDI-TOF.
5. Figure 3 was revised to correct mistakes.
6. In the method section, the description of MS/MS and MALDI-TOF sequencing was revised to include our new calibration experiments.
7. In the supplementary information, all data related to the sequencing were fully revised.
8. In the supplementraty information, Figure 13–41 were added.

Comment by comment responses.

We thank all reviewers for their elaborated and insightful comments on our manuscript. We fully accepted the major criticisms raised by the reviewer, and revised our manuscript accordingly. We reexamined all mass results and sequencing results, and revised our experimental section and supplementary information. We feel that we are greatly benefited from this revision, and, now, our manuscript is much improved from the original version. Our detailed responses to reviewers' comments are listed below.

Reviewer #1 (Remarks to the Author):

This manuscript reports a relevant synthesis approach to large sequence-defined polymers, and would nicely contribute to the efforts currently devoted to this new area of research. However, prior to being considered for publication in Nature Communications, it requires extensive revision in order to address the major concerns listed below.

In the proposed synthesis method, four dyad-containing monomers were first prepared by using different combinations of phenyllactic acid (P, defined as the 0-bit) and lactic acid (L defined as the 1-bit), hence yielding 00, 01, 10 and 11 coded dyads. Using a cross-convergent method, these dyads were then employed to prepare different reagents containing 4 bits of information i.e., tetrads), further reacted together to lead to byte-containing macro-monomers (i.e., reagents composed of 8 co-monomers) that were ultimately used to prepare long so-called PcL chains. As a result, these chains have a DP which is a multiple of 8, well suited to use the ASCII code to write information. Instead of using a (very) large library of reagents, this approach relies on the initial synthesis of the appropriate series of byte-containing macro-monomers needed for the preparation of any targeted final sequence, which indeed minimized the number of coupling steps to prepare long sequence-defined polymers.

At this point, the authors should better define the size limit they used to distinguish (co)oligomers from (co)polymers. In the abstract (line 3) as well as in the Introduction (page 2, second paragraph), they wrote that synthesis of sequence-defined species has been so far limited to oligomers and claimed that their method enabled preparation of polymers, the first example being chains of DP 64. However, sequenced-defined chains in this size range (as well as above) have already been reported a few years ago (e.g., works from the Lutz group reported in reference 18 as well as in Nat. Commun. 2017, 8, 967 which was omitted).

Our response

For information storage in synthetic polymers, we believed that one of the most important requirements is to achieve a long chain length without losing the ability to define aperiodic sequence in a scalable manner. Although solid-phase synthesis provides a reliable means to synthesize such polymers, high molecular-weight polymers (> 100 repeating units) having defined-sequences have been synthesized only in few occasions (notably by Lutz and coworkers). Our motivation for this work is to overcome the difficulties associated with solid-phase synthesis of high molecular weight polymers and to establish a synthetic route to

high molecular-weight polymers without statistical uncertainty in the number of repeating units and the sequence of monomers. In addition to the development of alternative media for information storage, we wish that our work could contribute to the ongoing efforts to the synthesis and study of monodisperse or discrete polymers that can provide new insights to understand how truly monodisperse and sequence-defined synthetic polymers behave. Based on these understandings, new possibilities of using these polymers for so far unforeseen applications could emerge.

To achieve our goal, we utilized the convergent growth approach to monodisperse polymers because the exponential growth of molecular weights of polymers can be achieved without relying on solid-phase synthesis using excess reagents. Previously, Hawker and coworkers pioneered this strategy to synthesize monodisperse polyesters with up to 64 repeating units of L-lactic acid or ϵ -caprolactone (Refs. 33 and 35). In these works, flash column chromatography was used as a purification method. From our investigation, we realized that polylactides having 64 or more repeating units could not be purified by conventional column chromatography due to the diminishing difference of affinity of products and reagents toward silica as a stationary phase. We reasoned that the limit of the convergent method for the synthesis of monodisperse polymers is an absence of purification methods to isolate the high molecular weight polymers from the reagents having a half the molecular weight of a product. Therefore, we adopted the size-exclusion chromatography as our purification method. This method allowed us to purify the desired high-molecular-weight from the reagents because the hydrodynamic volume of the coupled product is two times larger than the value of the reagents. By this method, we could purify polylactide and poly(phenyllactide) with up to 512 repeating units (\sim 38 kDa) in a gram scale.

One remaining challenge was a method to define aperiodic sequence by the convergent method. Due to the self-iterative nature, the convergent method to synthesize polymers has only yielded the polymers with repetitive sequences. We overcome this challenge by devising the cross-convergent method using permutations of monomers as the building blocks for the convergent growth. In case of using four dyads, any binary sequence can be constructed by the cross-convergent method, which can be directly translated to a binary digital code. We demonstrated that our method can be used for digital information storage in polymers.

With an expectation to distinguish oligomers and polymers in terms of their melt behavior, we studied the rheological behavior of monodisperse polylactides. Compared to the rheological results obtained from conventional amorphous PLAs, monodisperse PLAs behave as polymers by following a universal trend of the proportionality of melt viscosity to

molecular weight when the number of repeating units is 128 or greater. Based on the rheological behavior in bulk, we propose that monodisperse PLA could be considered as polymers when the number of a repeating unit is 100 or greater. We note that this range of molecular weight has not been achieved from synthetic monodisperse polymers, and, consequently, the physical behaviors of high molecular-weight monodisperse polymers has not been examined.

Previous results reported by Lutz and coworkers clearly meet these criteria of the number of repeating units or molecular weight. In our work, we limited the number of repeating units of sequence-defined PcL to 128 because of the limitation of our sequencing methods based on a tandem mass spectrometry and MALDI-TOF mass. As we demonstrated with polylactide and poly(phenyllactide)s having up to 512 repeating (~ 38 kDa), the number of repeating units of information-storing copolyester can be extended to 256 without difficulty if lactic acid and glycolic acid were used as monomers. We only demonstrated the synthesis of PcL with 128 repeating units due to the inability to read information by mass spectrometry.

With the comments from the reviewer in mind, we revised our manuscript to define the distinction between oligomers and polymers (page 2, second paragraph in the manuscript). The section introducing the previous works by Lutz and coworkers on the synthesis of sequence-defined polymers been revised. We also added a new paper by Lutz and coworkers in the reference section (Refs. 14). We appreciate the reviewer for drawing our attention to this valuable literature.

Another clear advantage of this cross-convergent approach was that any unreacted macro-reagent can readily be separated from targeted polymers since the mass of the latter ones increased exponentially. As a result, raw samples could easily be purified to yield highly monodisperse samples. This should be better acknowledged by the authors. First, the verb “synthesize” should be replaced by “obtain” in the sentence (page 5) “This purification method allowed us to synthesize monodisperse PcL in high yield ...”. Second, caption of Fig. 2b-d should be clarified: were data in Fig. 2b obtained for a raw sample prior purification and mass data shown in Figs. 2c-d recorded after purification of this sample?

Our response

As we described in our previous response, the size-exclusion chromatography warrants the purification of the desired products without the limit of the molecular weight of polymers. Because this method only relies on the difference of the hydrodynamic volume of a polymer for separation, the SEC purification could also be used to isolate the topological polymers such as cyclic polymers from their linear precursors. In our ongoing research, we confirmed that completely pure cyclic polymers could be isolated after the intramolecular cyclization of

linear monodisperse polymers because of the SEC-based purification and the absence of size distribution of linear polymer precursors. The versatility of SEC-based separation of monodisperse polymers is currently investigated, which will be published elsewhere. As the reviewer suggested, we revised our manuscript (page 5, last paragraph).

Figure 2b is a panel combining all normalized mass spectra of purified PcLs, which was composed to indicate the purity of PcLs in a wide range of molecular weight. We apologize for the confusion created by combining all mass spectra in a single figure. To avoid confusion, we revised the figure caption to indicate that each mass peak corresponds to the purified PcL. Figure 2c shows that the high molecular-weight PcLs (64-bit and 128-bit) do not contain residual fragments used for coupling and the product has no deletion error (missing residues at the terminus) due to the nature of the convergent growth approach.

Mass spectrometry was employed here as a characterization technique, using MALDI to generate sodium adducts of PcL in the gas phase. Surprisingly, according to the experimental section, no sodium was supplemented to the MALDI samples: this could have helped promoting better ionization of the largest chains and hence improve their MS/MS sequence coverage.

Our response

Under our experimental condition, observed mass peaks are predominantly those of sodium and potassium adducts as the molecular weight of a polymer increases. We presumed that metal cations originated from the solvent (THF, ACS grade) we used for our experiments. We observed similar results routinely when we measured MALDI-TOF mass with oligomers and polymers using organic solvents (ACS grade). In case of adding the Na source (sodium trifluoroacetate in acetone) in the matrix/polymer mixture in THF, we did not observe any enhancement of ionization of high molecular weight PAHs.

Moreover, low measurement accuracy achieved when using MALDI-MS(/MS) does not permit to report m/z values with the precision provided in Supplementary Tables 1-15: m/z values may be reported with one decimal digit (certainly not two). Moreover, it is not clear whether unit resolution was achieved for all ions and whether ions were measured at their monoisotopic values. For example, in Tables 7-9, while the $[M+Na]^+$ species is expected at a monoisotopic m/z value of 2081.8, its maximum isotopic value is m/z 2082.8: according to the reported experimental m/z of 2082.3, one can either suspect that resolution was too low to distinguish isotopes or that there was an issue with mass calibration. Similar issues are encountered in Table 10 (isotopic maximum calculated at m/z 2158.8).

Our response

Concerning the general method of mass determination and the analysis of data:

We thank the reviewer for the detailed comments concerning on the mass experiments and the handling of data. We feel that we are privileged for having these comments from the reviewer. We remeasured the mass values of all compounds used in this study, and revised the manuscript following the reviewer's detailed comments. We fully accept the reviewer's comments and suggestions, and revised our manuscript and supplementary information to correct the mistakes. Throughout the manuscript, we used the isotopic maximum m/z as the calculated mass of our polymers and fragments with molecular weight under 10 kDa. In a case of polymers with molecular weight greater than 10 kDa, we used the molecular mass as the calculated mass value.

We recalibrated the instrument with new calibration kits (ProteoMass peptide/ protein MALDI-MS calibration kits, Sigma). With the reviewer's criticisms in mind, we remeasured all compounds and polymers used in our study with MALDI-TOF mass or MALDI-TOF MS/MS. With new calibrations, we confirmed that the experimental m/z values of the compounds correspond to the calculated values (isotopic maximum m/z) with an acceptable accuracy.

For reporting the measured mass values, we reported all m/z values obtained from MALDI-MS(/MS) with one decimal digit. We found that MALDI-TOF spectra of polymers with less than 3 kDa showed peaks that can be accurately assigned to the isotopic maximum value. The experimental values slightly deviated from the theoretical mass when the molecular weight of the polymer is ~ 7 kDa or greater (Figure S1). The deviation from the theoretical mass increased as the molecular weight of PcL increased.

Figure S1. MALDI-TOF mass spectrum of a PcL, SEQUENCE (theoretical isotopic maximum mass: 7672.7 Da)

The mass values for the Supplementary Tables 1–6 were corrected with new experimental mass values obtained by MALDI-TOF MS/MS. In a similar manner to the MALDI-TOF experiments, the measured values by MS/MS were well coincided with the theoretical values except the samples with high molecular weight.

For PcLs with higher molecular weights, the deviation of the measured mass from the theoretical value becomes noticeable. For example, the isotopic masses of the PcL (16-bit, SE, Supplementary table 7) with sodium (isotopic maximum mass 2082.8 Da) were found at 2081.9, 2082.9, 2083.9, 2084.9 and 2085.9 from MALDI-TOF (Figure S2). We used the peak with a maximum intensity, which corresponded to the theoretical isotopic maximum m/z .

Figure S2. MALDI-TOF mass spectrum of the PcL (16-bit, SE)

In MALDI-MS/MS spectrum, we could detect isotopes of $[SE+Na]^+$ ions: 2081.9, 2083.0, 2084.2, 2085.3, 2086.5 and 2087.6. The experimental m/z of this molecule is 2083.0 Da (Figure S3).

Figure S3. MALDI-TOF MS/MS spectrum of the PcL (16-bit, SE)

Similar results were obtained from 16-bit long PcLs as shown below.

PcLs, QU and EN (isotopic maximum m/z 2082.9 Da in MALDI-TOF) showed experimental m/z 2083.0 in tandem mass experiments (Figure S4).

Figure S4. MALDI-TOF MS/MS spectra of the PcLs (16-bit, QU (left), EN (right))

PcL (16-bit, CE) has m/z 2159.0 and m/z 2159.1 (at isotopic maximum) in MALDI-TOF MS and MALDI-MS/MS spectrum, respectively (Figure S5).

Figure S5. MALDI-TOF mass spectrum (left) and MS/MS spectrum of the PcLs (16-bit, CE)

In contrast, calculated values reported for $[M+Na]^+$ in Tables 11 and 12 are wrong. In Table 11, this should be m/z 3918.4 (monoisotopic) or m/z 3920.4 (at isotopic maximum). In Table 12, this should be m/z 3994.4 (monoisotopic) or m/z 3996.4 (at isotopic maximum). With this regard, data of all Supplementary Tables have to be thoroughly revised.

Our response

We apologize our mistakes in using the miscalculated values for the tables 11 and 12. We thank again the reviewer for correcting our mistakes.

For Supplementary Table 11, PcL (SEQU, theoretical isotopic maximum mass: 3920.4 Da) showed the isotopes from MALDI-MS/MS spectrum: 3918.4, 3915.5, 3920.7, 3921.8, 3923.0 and 3924.1 (Figure S6). The m/z (isotopic maximum value) of this compound is 3920.7 Da.

Figure S6. MALDI-TOF MS/MS spectrum of the PcL (32-bit, SEQU)

For Supplementary Table 12, PcL (ENCE, 3996.4 Da) compound is composed of isotopes in MALDI-MS/MS spectrum: 3994.4, 3995.6, 3996.8, 3997.9, 3999.0 and 4000.2 (Figure S7). The m/z (isotopic maximum value) of ENCE is 3996.8 Da.

Figure S7. MALDI-TOF MS/MS spectrum of the PcL (32-bit, ENCE)

The discrepancy of the observed mass from the calculated value in MS/MS results is originated from the process of MALDI-MS/MS. For the analysis of fragments, we use the monoisotopic m/z as a parent ion. As the molecular weight of PcL increases, the detected mass value deviates from the calculated monoisotopic m/z value. This discrepancy distorts the proceeding mass values slightly.

In contrast, MALDI-TOF mass showed the peaks well coincided with the calculated mass values. We could observe the isotopes of PcL (32-bit, SEQU) in a MALDI-TOF MS spectrum at 3918.4, 3919.4, 3920.4, 3921.4, 3922.4 and 3923.4 (Figure S8). The monoisotopic and isotopic maximum m/z is 3918.4 and 3920.4, respectively.

Figure S8. MALDI-TOF MS spectrum of the PcL (32-bit, SEQU)

Similar result was obtained from PcL (32-bit, ENCE) (Figure S9). Experimental m/z of isotopes were well agreed with the theoretical values: 3994.5, 3995.5, 3996.5, 3997.5, 3998.5, 3999.5 and 4000.4 (calculated mass value is 3996.4 Da).

Figure S9. MALDI-TOF MS spectrum of the PcL (32-bit, ENCE)

Similar remarks apply for m/z values reported for species shown in Figure 5, particularly those measured in the TOF linear mode that exhibit very large error. Moreover, data in the caption of this figure are very misleading as it appeared that m/z values were sometimes mixed with mass values.

Our response

As the reviewer instructed, we remeasured all PAH samples with a recalibrated mass spectrometer. The manuscript and the supplementary information were revised with these new results. We also used the correct terms to indicate our experimentally found m/z values. For small PAHs, we found experimental values that agreed with the calculated values. However, the results of higher molecular weight PAHs (> 10 kDa) after recalibration could not be well resolved with an equivalent precision of the peaks of the polymers with 64 or smaller repeating units.

In MALDI-TOF MS spectrum of LA128 (theoretical isotopic maximum mass of $[M + Na]^+$: 9468 Da), isotopes could still be distinguished in spite of peak broadness (Figure S10).

Figure S10. MALDI-TOF MS spectrum of LA128

PAHs with the molecular weight higher than 12 kDa showed significantly broad peaks by MALDI-TOF mass in the reflection mode. MALDI-TOF spectrum of PA80 (theoretical molecular mass: 12098 Da) showed a merged peak (Figure S11). Identification of isotopic mass values is no longer possible in this molecular weight region.

Figure S11. MALDI-TOF MS spectrum of PA80

PAHs with high molecular weight (> 20 kDa) only can be investigated with MALDI-TOF in the linear mode. Linear mode measurement only showed broad peaks around the expected

mass. The difference between the calculated and experimental mass values is much larger than that observed from MALDI-TOF in the reflectance mode. For this reason, we did not use the linear mode mass for sequencing of the polymer. Significant peak broadness observed from the linear mode MALDI-TOF spectra was found routinely in previous results on dendrimers, discrete oligomers, and proteins. In spite of the peak broadness, MALDI-TOF results, combined with GPC results, confirmed the presence of high molecular weight PAHs without contamination (Figure S12).

Figure S12. GPC (a) and MALDI-TOF mass (b) results of LA512

Because the two coded bits had different masses (P: 148 Da, L: 72 Da), MS/MS could be employed as a sequencing method. A nomenclature is available to designate MS/MS fragments of synthetic polymers (see Wesdemiotis et al *Mass Spectrom. Rev.* 2011, 30, 523) and should be employed here. According to this nomenclature and owing to the bond cleavage experienced by PcL, fragments containing the left-hand side (alpha) termination should be named a_i while fragments containing the right-hand side (omega) termination should be named y_j (with i and j the number of monomers they contain, respectively). Accordingly, analysis of a_i fragments should allow the sequence chain to be partially reconstructed from the left to the right, whereas y_j permitted to partially decipher 0/1 bits from the right to the left. This partial sequence coverage is not really acknowledged in the dissociating scheme shown in Figure 3a. It also means that both fragment series need to be considered to recover the whole polymer sequence, even for quite small species containing one byte of information.

Details reported in the experimental section are not sufficient to figure out how MALDI ions were activated and whether the activation energy was increased with the polymer DP. In addition, data reported in Supplementary Tables are confusing in terms of sequencing. For the sake of simplicity, let's take the example of Supplementary Table 1 reporting MS/MS data recorded for the 01010011 chain (coding for S). First, cleavage of the C-O bond should proceed according to a rearrangement which leads to the transfer of one proton from the

right- to the left-hand of the dissociation bond, resulting in i) a_i fragments that contain the original alpha group and a new OH termination and b) y_i fragments that contain the original omega group and a new termination which depends on the nature of the unit before which cleavage occurred: $-\text{CH}=\text{CH}_2$ if cleavage occurred before L or $-\text{CH}=\text{CH}_2\text{-Ph}$ if cleavage occurred before P. Detection of one or the other fragment series depends on which part of the cleaved chain is the adducted sodium. This is not “bidirectional fragmentation” but dissociation that leads to complementary fragments. On the one hand, fragments annotated in orange at m/z 523, 595, 743, 891 and 963 respectively correspond to a_4 - a_8 fragments (calculated m/z values are false) and m/z difference measured between these ions allowed the five last units to be identified as L, P, P, L and L. Again, these fragments contain the original alpha end-group and allow this partial sequence to be re-constructed from the left- to the right-hand side. On the other hand, fragments annotated in blue at m/z 405, 553, 625, 773 and 845 respectively correspond to y_3 - y_7 fragments (calculated m/z values are false) and m/z difference measured between these ions allowed the five first units of the original chain to be identified as P, L, P, L and P (the first P unit being identified from the mass difference between $[\text{M}+\text{Na}]^+$ and y_7). In summary, in great contrast to annotations found in Supplementary Table 10, blue ions contain the original omega termination and allow the sequence to be reconstructed from the right to the left. This aspect has also to be corrected in all Supplementary Tables as well as in explanations provided for sequencing in the main text. However, this fragmentation pattern did not allow reliable sequencing of the longest chains since there is no part of the sequence covered by both a_i and y_j fragments.

Our response

We deeply appreciate detailed comments, criticism, and suggestions raised by the reviewers. We reanalyzed our MS/MS results according to the suggested reference (Ref. 46 in our revised manuscript), and revised all tandem mass analysis and sequencing (Supplementary Table 1–13). We revised our method section and supplementary information to reveal all details of our experimental conditions.

According to the reference, we changed the notation for fragmentation patterns and assigned the fragments appeared in the tandem mass spectra accordingly. We designated all fragments including terminal fragments using the notation of a series of y_i fragments (y_1 , y_2 ,...) and a_i fragments (a_2 , a_3 ...) (representative figure shown below). For sequencing of PcL, we excluded mass peaks of terminal fragments due to their low peak intensity.

Figure S13. MADLI-MS/MS spectrum with peak assignment of PcL (8-bit, S)

S (PLPLPPLL)									
Si → Bz	Theoretical exact mass (molecular mass)	Found m/z	Difference	Sequence	Bz → Si	Theoretical exact mass (molecular mass)	Found m/z	Difference	Sequence
[M+Na] ⁺	1125.4 (1126.3)	1125.5			[M+Na] ⁺	1125.4 (1126.3)	1125.5		
y7	845.3 (845.9)	845.5	280.0	Si-P	a8	963.4 (964.1)	963.5	162.0	Bz-L
y6	773.3 (773.8)	773.5	72.0	L	a7	891.3 (892.0)	891.5	72.0	L
y5	625.2 (625.6)	625.5	148.0	P	a6	743.3 (743.9)	743.5	148.0	P
y4	553.2 (553.6)	553.4	72.1	L	a5	595.2 (595.7)	595.5	148.0	P
y3	405.1 (405.4)	405.3	148.1	P	a4	523.2 (523.7)	523.5	72.0	L
y2	257.1 (257.3)	257.3	148.0	P	a3	375.2 (375.5)	375.4	148.1	P
y1	185.1 (185.2)	185.2	72.1	L	a2	303.1 (303.4)	303.4	72.0	L

Table S1. Decoding table of PcL (8-bit, S)

For high molecular weight PcL (64-bit, SEQUENCE), we observed the discrepancy between the calculated and experimental mass values from the tandem mass experiment (Supplementary Table 12). This was caused primarily from the deterioration of the resolution of MALDI-TOF, which causes the peak broadness and inaccurate picking of the parent molecular ion used as a reference for the assignment of the mass of fragments. For 64-bit PcL, MALDI-TOF mass could not identify the monoisotopic mass peak accurately due to the low intensity and broadness of observed peaks (Figure S14).

Figure S14. MALDI-TOF MS spectrum of PcL (64-bit, SEQUENCE). Experimental isotopic maximum m/z 7674.5 Da and expected experimental monoisotopic m/z 7669.5 Da.

For reading the sequence of PcL, we use the mass difference of adjacent fragments. We could read the mass difference of two monomers (72 and 148 Da) within an accuracy of less than 2 Da difference regardless of the discrepancy observed from the tandem mass spectrum of 64-bit PcL.

The authors wrote that the upper DP for reliable sequencing was 90 but reported data provided evidence for sequencing of chains only up to DP 64. As a result, the polymer with DP 128 had to be chemically degraded and the implemented hydrolysis reaction was shown to proceed from the alpha to the omega termination, yielding a series of products that differ from one another by the mass of a single unit that could be identified in MS. Moreover, MS/MS still had to be performed to identify the last byte of information. Although very interesting from the analytical point of view, this directional hydrolysis pathway was neither explained nor rationalized. Still, the need for performing such a chemical degradation to read messages of more than 64 bits remains a main drawback for these PcL polymers. In other words, this manuscript does report easy sequenceability for polymers of larger size than other species reported in the literature. In this context, the last sentence of the abstract which deals with a perspective not demonstrated in the present manuscript should hence be removed; instead, the authors should acknowledge that full MS/MS sequence coverage could only be demonstrated up to DP 64 and that a sample pre-treatment had to be performed for partially selective cleavage of longer chains prior subjecting so-obtained products to MS and MS/MS analysis. The last part of the manuscript deals with the synthesis of even larger polymers but

appears somehow out of scope of the study since this is no longer about sequence-defined species but homopolymers made of P or L units and study of some of their macroscopic properties.

Our response

Large scale information storage requires the distribution of data into many polymer chains. This mandates that each polymer chain has to store the address data and the raw data, which is required to retrieve full information from the sequencing of multiple polymer chains. For this reason, we believed that the chain length of PcL is a crucial factor for its application in data storage. For example, 64-bit PcL only can store 32-bit of raw data if the address consumes 32 bits. Our motivation for this work is to devise a synthetic method to prepare sequence-defined polymers with a large number of repeating units, which can store enough amount of bit information for distributed storage.

Although the tandem mass technique is widely used for the sequencing of proteins and polymers, there exists the limit of the molecular weight of the parent molecular ion used for fragmentation. In our case, PcL with 64 repeating units could be reproducibly sequenced by a MALDI-TOF MS/MS, but PcLs with more repeating units could not be fully sequenced. Higher molecular weight PcL did not show any signal in tandem mass measurements.

Aliphatic polyesters such as poly(α -hydroxy acids) can readily be degraded to their repeating units via hydrolysis of ester bonds, which makes them degradable alternatives to olefin-based polymers in commodity applications. In our current research, we specifically intended to use (bio)degradable polymers as media for information storage.

The chemical degradation of copolyester in a basic condition would give randomly degraded fragments via base-catalyzed hydrolysis of ester groups in the polymer backbone. The resulting mixture of the degraded polyester could be directly used for conventional MALDI-TOF mass, which provided a series of mass peaks representing all fragments originating from the parent polymer.

Upon hydrolysis, PcL is dissociated into two fragments. One is the *ci* fragment that contains the original alpha group and a new carboxylic acid terminus. The other is the *xi* fragment that contains the original omega group and a new hydroxyl terminus. For decoding the sequence, we used the peaks corresponding the series of *xi* fragments because of their high intensities compared to the peaks of *ci* fragments.

This is one example of hydrolysis of PcL (16 byte SEQUENCESEQUENCE), which is fragmented into x90, containing hydroxyl and benzyl ester terminals, and c38, containing carboxylic acid and TBDMS terminal, fragments. x90 fragment is observed in MALDI-TOF spectrum (Figure S15).

Figure S15. MALDI-TOF MS spectrum showing the mass of x90 fragment.

The experimental mass values of the fragments were well matched to the calculated values. We did not observe any deletion error from the high molecular weight fragments. For sequencing, we used the mass peaks of x_i fragments to read the mass difference between adjacent fragments with an accuracy within 3 Da.

The sequencing by MALDI-TOF mass of the degraded PcL could be carried out by a single set of measurements. The mass spectrum was acquired by a single measurement in a range of 1000 Da to 16000 Da. For the last 8 residues (> 1000 Da), we used a MS/MS for the sequencing because of the noise arising from the matrix molecules used for MALDI-TOF mass.

The full sequencing of the synthetic polymer with 128-repeating unit has not been realized previously by any other method.

We choose a fragment x22 as a parent ion, which is analyzed by MALDI-TOF MS/MS. The fragmentation of x22 was given below.

Mass of a-series fragments corresponded to m/z values in MALDI-TOF MS/MS spectrum (Figure S16).

Figure S16. MALDI-TOF MS/MS of the x22 fragment.

Although the degradation of polymers under mild conditions could not be applied universally, we believe that the MALDI-TOF mass of polyesters after chemical or biological degradation could be a facile method for the sequencing of these polymers with a large number of repeating units. For sequencing, we read the mass difference between two successive fragments. We confirmed that MALDI-TOF mass operating in the reflectance mode provides a resolution sufficient to read the molecular weight difference of 14 Da. The limit of the molecular weight of the analyte is ~ 20 kDa. With a 20 kDa limit, we could decipher 256-bit information encoded in the copolyesters composed of L-lactic acid and glycolic acid (mass difference is 14 Da). We are currently working on the enzymatic degradation of copolyester of L-lactic acid and glycolic acid, and the subsequent sequencing by MALDI-TOF. This work will be published elsewhere.

The section concerning the scalability of the convergent method to synthesize poly(α -hydroxy acid) (PAH) was added to this paper because our goal is to establish a synthetic method to prepare PAHs with a large number of repeating units. We synthesized poly(rac-

lactide) with 512 lactic acids. We did not synthesize sequence-defined polymers for this part because the limitation of our sequencing method. We could not read the sequence of a PcL with 256 repeating units (30 KDa) because of the inability to use the reflectance mode for high molecular weight samples in MALDI-TOF experiments. In addition, we wish to measure the rheological properties of monodisperse polymers, and to compare the measured values to the existing results obtained from conventional polylactides. By measuring the zero-shear viscosity of monodisperse PLAs, we confirmed that the monodisperse PLA followed a universal behavior of polymers, which applies to all polymers regardless of their backbone structures. In terms of thermal behaviors, we concluded that monodisperse PLAs with more than 100 repeating units behave like conventional polymers, for which the molecular weight is a property-deciding factor.

As reviewer suggested, we added a new paragraph to describe the background and details of the degradative sequencing of PcL with MALDI-TOF (page 9). We also take the reviewer's criticism concerning the concluding sentence of our abstract. We revised our abstract accordingly.

Reviewer #2 (Remarks to the Author):

The manuscript describes a convergent approach to sequence-encoded macromolecules via esterification reactions. Monodisperse macromolecules with up to 512 repeat units are synthesized, and a molecule with the binary code for the word "sequence" is presented. While the present approach by itself might be novel, and surely a lot of work was dedicated to reach this aim, I am a bit afraid that the approach is not overly novel. The binary system in use has already been introduced by Lutz and coworkers several years ago, with similar coding lengths being described. The present chemistry requires protecting group chemistry (even if it is a simple one), while other groups (e.g. Du Prez or Meier) have shown that orthogonal protecting-group free approaches are largely superior. Also the idea to synthesize smaller fragments that are then coupled later has been shown (qr codes by Du Prez) before. Characterisation of the sequences, as much as the determination of physical parameters of the discrete materials follows literature examples and delivers nothing beyond the already known (the authors might also want to check De Neve, *Angewandte Chemie*, 2019 for further literature). In other words, I think this is good work, and I have no real detailed comments at this stage (which doesn't mean there wouldn't be any), but don't think that this manuscript is really novel. Neither the approach is new, nor the chemical system or the analysis of the polymers is.

Our response

Scalable synthesis of polymers composed of a large number of repeating units arranged in specific sequences has been a long-pursued goal of polymer chemistry. Traditionally, polymers with absolutely defined molecular weights and sequences have been synthesized almost exclusively by the sequential additions of monomers via solid phase synthesis. In recent years, non-biological sequence-defined polymers with a large number of monomers (> 100 units) have been realized via step-wise addition of monomers (Refs. 13). This method confers a perfect degree of freedom in terms of sequence definition of the resulting polymer. However, few drawbacks, in our opinion, remain for the synthesis of high molecular weight and sequence-defined polymers by solid-phase synthesis. When the target molecular weight (degree of polymerization) is high, the required number of coupling steps increases proportionally. The increased number of coupling steps demands the use of excess reagents for the completion of the reactions and repeated purification steps to ensure the error-free encoding of sequences for high molecular weight polymers. In addition, the confinement of the growing polymer chains on the surface of the solid support imposes the steric restriction for further growth when the molecular weight of the growing chain reaches a critical value.

This increases the possibility of having incomplete coupling at the chain terminus, which potentially causes deletion errors. These drawbacks might hinder the availability of synthetic polymers with unambiguously defined chemical structures in a large scale, which will be required to investigate the potential of these polymers as new materials.

The convergent synthetic approach has been used as an alternative to solid-phase synthesis for obtaining perfectly-defined macromolecules such as proteins, nucleic acids, dendrimers, and polymers. This method is beneficial for achieving high molecular weights of synthesized macromolecules as each iteration of the convergent coupling steps doubles the number of repeating units of the resulting polymer and warrants the exponential growth of its molecular weight. These characteristics enabled us to synthesize high molecular weight polymers without molecular weight distribution with a reduced number of synthetic steps (compared to step-wise additions by solid phase synthesis) without using a large excess of reagents. In spite of this advantage, a significant challenge has remained unaddressed: the self-iterative nature of the convergent growth approach has only allowed the polymers having repetitive sequences. Therefore, the encoding of aperiodic sequences by the convergent method has not been realized.

Our motivations for the current work are (1) to synthesize polymers without statistical uncertainty in molecular weights and sequences via a convergent method that is scalable in molecular weight and quantity; (2) to establish an efficient method to define random sequences of monodisperse polymers built by the convergent method.

Concerning our cross-convergent method for encoding binary information:

The reviewer pointed out the previous report by Lutz and coworkers concerning the use of constituent building block storing multi-bit information (*ACS Macro Lett.* **2019**, 8, 1002). This work used a one-to-one correspondence between permutations of two-bit information (00, 01, 10, 11) to four chemically distinct monomers, which were introduced to form oligomers and polymers via a sequential step-wise coupling. This direct correspondence of multi-bit information to monomers or building blocks has been utilized in other recent reports (De Prez et al. *Nat. Commun.* **2018**, 9 4451 (Refs. 19); Whitesides et al. *ACS Cent. Sci.* **2019**, 5, 911. (Refs. 25))

For our work, we also used four dyads of two monomers to construct an aperiodic binary sequence. We translated given binary information using four dyads, which could be constructed as a series of limited numbers of 4-bit long sections (tetrads). This translation allowed us to build copolymers containing binary codes by the convergent growth method.

We coined the term “cross-convergent method” to indicate our strategy to build sequence-defined polymers via a convergent approach. We believe that our result can be differentiated from the solid-phase synthesis of sequence-defined polymers in two aspects: (1) the molecular weight of the resulting polymers could be scaled, and the quantity of the product could be increased to a multi-gram scale. (2) polymers with aperiodic sequences could be synthesized by a minimal number of steps with stoichiometric amounts of reagents. We also stress that our copolyesters (PcLs) could store up to 50% more data than DNA (per Da basis) due to the simple chemical structures of repeating units. Therefore, these sequence-defined copolyesters could be low-cost alternatives to DNA for large-scale information storage.

Concerning the chemistry employed to synthesize polymers:

The convergent method to grow macromolecules has been widely used for the synthesis of dendrimers and polymers. We chose α -hydroxy acid as a monomer because the convergent synthetic strategy of α -hydroxy acid to monodisperse oligomers and polymers has been established by pioneering works by Hawker and coworkers (Refs. 33 and 35). Recently, the Meyer group and the Meijer group used this strategy to investigate the impact of monodisperse and sequence defined polymers on the degradability and the self-assembly of the resulting polymers and block copolymers (Refs. 44; *Angew. Chem. Int. Ed.* **2019**, *58*, 10747 (Refs. 57)). In addition, the resulting poly(α -hydroxy acid)s offer the opportunities to investigate the physical properties of monodisperse version of commonly used polymers such as polylactides, which could provide the chances to discover new and distinctive properties of monodisperse and sequence-defined polymers that could be obscured by the statistical distribution of sizes of conventional polymers. Therefore, we adopted this well-established chemistry utilizing orthogonal protection/deprotection to synthesize poly(α -hydroxy acid) with a large number of repeating units (> 512 monomers), which has not been realized in previous studies.

The coupling chemistry employed for the synthesis of discrete oligomers and polymers has to ensure the occurrence of only a single reaction between functional groups to prevent uncontrolled polymerization. Therefore, the coupling chemistry requires the regulation of the reactivity of functional groups in a bimodal fashion (on and off). Orthogonal protection has been commonly utilized to achieve the on-off regulation of functional groups. As the reviewer suggested, recent results reported by Du Prez and Meier utilized novel coupling reactions such as multicomponent reactions to construct sequence-defined polymers by stepwise additions of monomers. As the reviewer pointed out, these results avoided the

reliance on the protection strategy to regulate the reactivity of functional groups. However, these examples also require additional steps for the activation of functional groups in dormant states, such as a reduction step for Passerini reaction to expose carboxylic acid and a ring-opening reaction for thiolactone to generate thiol for Michael addition. In addition, these chemistries have not been applied for the convergent synthesis of sequence-defined polymers with large number of repeating units. We completely agree with the reviewer's opinion that the examples suggested might be vastly superior to the existing methods based on protection or deprotection chemistries if the requirement of the additional activation steps could be lifted and the simplified backbone structures analogous to common polymers could be realized. An elegant chemistry reported by Barner-Kwolik and coworkers (*Angew. Chem. Int. Ed.* **2019**, *58*, 7133 (Refs. 26)) suggests that this goal could be achieved. We are working diligently to find a solution on these issues, which will bring the significant improvement on the atom and step economy of the synthesis of sequence-defined polymers having large numbers of repeating units.

Concerning the measurement of the physical properties of monodisperse polymers:

We thank the reviewer for guiding us to an important recent paper by Junkers and coworkers. We added this work in our reference section (Refs. 55). They demonstrated the measurement of selected physical properties of discrete oligomers (1~22 units of methyl acrylates) in solution. This remarkable paper stresses the significance of the synthesis of high molecular weight polymers without molecular weight distribution to investigate the properties of polymers without statistical uncertainty. The paper also emphasizes the significance of the monodisperse polymer having identical backbone structures to commonly used commercial polymers, which will allow the comparison of measured properties between monodisperse and polydisperse variants of the same polymers. Prof. Junkers and coworkers, however, only studied the solution properties of oligo methyl acrylates due to the difficulty of the purification of polydisperse oligomers to discrete oligomers by flash column chromatography. The study of physical properties of high molecular weight polymers that possess identical backbone structures of commonly used polymers remains unfulfilled.

In our paper, we detailed our purification method based on size-exclusion chromatography, which allowed us to obtain pure monodisperse polylactides with up to 512 repeating units in a gram-scale. Conventional polylactides and their physical properties in relation with their molecular weights have been extensively studied. Our results are the first examples of the measurement of bulk properties of monodisperse polylactides under chain entanglements.

Our results, therefore, could be a starting point of endeavor to understand the physical properties of monodisperse polymers having high molecular weights, which could serve a reference for optimization of the physical properties of polymers. Along with the pioneering results by Hawker, Meijer, Meyer, and Junkers, we believe that our results could contribute to the exciting research to discover the structure-property relationship of synthetic polymers without statistical uncertainty in molecular weights and sequences.

Reviewer #3 (Remarks to the Author):

Kim and coworkers show the synthesis of discrete and sequence defined phenylactic-co-lactic acid polymers using a cross-convergent method resulting in a nicely scalable method. Overall, this is a nice paper where they show how these discrete polymers with an aperiodic sequence can be used as for data storage, an ever growing field of interest. The results are promising as these polymers are relatively cheap (compared to DNA), even though the synthesis and purification can be time consuming. Hence, this work contributes towards solving the data storage problem in the future and is therefore of importance to a broad audience. The paper is recommended for publication after major revision, where the following points are addressed in a convincing way.

A structure-property relationship for the high Mw lactic acid polymers is shown. This is interesting, though, it would be more interesting to show this for the copolymers. The manuscript is focused on the copolymers and therefore it is much more important to show the structure-property relationship of the copolymers. Then, the question on the importance of the sequence is shown not only for data storage but also for polymer properties. Showing only the rheological measurements of the LA polymers does also not fit in the flow of the manuscript. One could then answer the question of on what the effect is of the exact sequence on the polymer properties. Would you be able to also perform the rheological measurements on the copolymers and elaborate on the importance of the exact sequence for polymer properties?

Our response

We thank the reviewer for pointing out the important aspect of the relationship between physical properties and sequences of discrete polymers. In this work, we wish to show that the convergent growth method, in conjunction with size exclusion chromatography, could prepare monodisperse and sequence-defined polyesters in a scalable manner in both molecular weight and preparation quantity. We believe that a large scale (at least a multi-gram scale) production of these polymers could provide a foundation to discover new properties of polymers without statistical uncertainty in molecular weights and sequences. As we described in this work, monodisperse amorphous polylactides behaved in a similar

manner to conventional polylactides when the number of repeating units is close to 100. In particular, the rheological measurement of zero-shear viscosity of a series of polylactides (LA128, LA256, and LA512) showed a linear relationship proportional to the 3.4th power of M , which coincides with a universal relationship of synthetic polymers. This universality is found from polymers regardless of their nature of chemistries. We expect that the rheological measurement of viscosity of sequence-defined PCLs would also show the universal agreement to the relationship, $\eta \approx M^{3.4}$. We interpret that this behavior of monodisperse polymers arise from the statistical nature of the chain entanglement. Our intention with this demonstration is to show that the monodisperse polymers follows a universal behavior of synthetic polymers in spite of their discrete molecular weights.

Concerning the reviewer's suggestion and criticism, we completely agree with the reviewer's view point of the importance of the defined sequence of polymers to their physical properties. We believe that monodisperse polymers should behave in different ways compared to conventional polymers if the observed physical properties do not rely on probabilistic incidents such as chain entanglements. For example, Meyer and coworker reported that the degradability of poly(lactic-co-glycolic acid) could be fine-tuned by defining the sequence of two monomers in the polymer backbone. We are currently studying the relationship between the sequence and the (bio)degradability of polyester prepared by the cross-convergent method. We are also studying how the absence of molecular weight distribution affects the behavior of block copolymers, PLA-b-PPAs.

In the second paragraph of the introduction, the authors describe ways to make discrete macromolecules or oligomers/polymers. They describe (cite) many examples of linear macromolecules synthesized via an iterative convergent method. However, the authors use the cross-convergent method and it would be beneficial to know a bit more about this method before explaining the synthesis.

Our response

The convergent growth method allows a rapid build-up of the molecular weight of monodisperse macromolecules. However, the self-iterative nature of this method prevents the realization of the polymer with a randomized sequence of monomers. This inability of the encoding of aperiodic sequences remains an unsolved problem for the convergent method. The cross-convergent method we described in this paper is a way to build a randomized sequence of two or more monomers utilizing the smallest units covering all possible permutations of the arrangements of monomers. For example, 2-bit system only requires four

permutations of monomers (00, 01, 10, 11) to express any random sequence. We translated the entire binary code to a series of dyads. The cross-convergent steps were used to form 4-bit words, which consequently were converged to form larger words and, essentially, fully grown sequence. Although these smallest units (dyads of monomers) can be successively linked to form a polymer by solid-phase synthesis, we decided to use the convergent method to express a random binary code to exploit the benefit of the convergent growth method. By using the cross-convergent method, any randomized sequence could be expressed without losing the benefits of the exponential growth of molecular weight and the minimized use of synthetic steps and reagents. Combining the combination of dyads to form a 4-bit segment and subsequent convergent steps to build the 8-bit word and the entire sequence, we coined a term 'cross-convergent method' to describe our approach to sequence-defined polymers. The detailed description of the cross-convergent method is described in the manuscript (page 4 and 5). To indicate this, we revised our manuscript (page 3, first paragraph).

The authors show that the synthesis of the polyester can be nicely performed in an iterative way. However, can the authors elaborate on the choice for racemic lactic acid and phenyllactic acid? Would similar results be obtained regarding data storage when only one of the two monomers was chosen, yet, controlling the stereochemistry?

Our response

Our motivation for this work is to establish a synthetic route to high molecular weight polymers without statistical uncertainty in the number of repeating units and the sequence of monomers. We wish that our work could contribute to the ongoing efforts to the synthesis and study of monodisperse or discrete polymers that can provide new insights to understand how the truly monodisperse and sequence-defined synthetic polymers behave. For this goal, we intentionally used racemic α -hydroxy acids for building polymers because of our interest in synthesizing amorphous polymers for studying their thermal behavior. As the reviewer suggested, stereospecific α -hydroxy acid can readily be used under the same preparation condition for sequence-defined polymers. Although binary information can be encoded as a sequence of L- and D-stereoisomers of the same α -hydroxy acid such as lactic acid, our current decoding method relies on a tandem mass spectrometry or MALDI-TOF mass spectrometry. The mass spectrometric sequencing requires the mass difference between bit-encoding monomers. Therefore, the stereochemical encoding of information necessitate new decoding methods for sequencing. Recent advancement of next-generation sequencing of

DNA could provide an inspiration for the novel methods of sequencing of our polymers storing information in a stereochemical manner.

What is the reason why PA could be synthesized up to 256 repeating units while the LA could be synthesized up to 512 repeating units in an efficient way?

Our response

The convergent growth of the polymer chain requires the coupling reaction between functional groups at the chain ends of macromolecular reagents. The coupling reaction between macromolecular precursors may experience the difficulty in exposing the chain end to the surrounding, which originates from the random coil conformation of a high molecular weight polymer. For PA and LA, this difficulty of finding the chain end arose when the molecular weight of the reagents exceeded ~ 37 kDa. In addition, the viscosity of the solution dissolving these high molecular weight reagents was substantially higher than that of the solution of lower molecular weight reagents, which required the dilution of the reagent concentration. The solution viscosity is also proportional to the molecular weight of the polymer. We reasoned that a significantly lower yield of the coupling reaction between high MW reagents was originated from these reasons.

To circumvent this problem and to extend the scope of our method, we are currently investigating the coupling reaction under the flow chemistry condition. We expect that the conformation of the polymer precursors could be stretched to expose the chain end under the shear flow within the capillary.

The rheological measurements of the LA polymers with 128 or more repeating units indicate the presence of entanglements in the bulk. Does this imply that all polymers with a lower molecular weight do not have any entanglements? Where is the point for having entanglements since the M_e was measured to be 3920 Da?

Our response

Polymer chains believed to have physical entanglements when their molecular weights exceed M_c (critical molecular weight, $M_c \approx 2M_e$). M_e refers the molecular weight between entanglements, which requires at least 2 points of entanglement. The critical molecular weight of chain entanglement (M_c) is the parameter at which given polymer experience significant chain entanglement. Under the M_c , polymer chains move as if they experience no chain entanglements ($\eta \approx M$). To avoid confusion, we added an explanation of M_e and M_c in our manuscript (page 13-14, first paragraph).

REVIEWERS' COMMENTS:

Reviewer #1 (Remarks to the Author):

With this revised version, the authors nicely addressed most concerns raised by the reviewers and their efforts to do so should be acknowledged.

However, the authors did not convincingly consider the issue which was raised by all three referees regarding the relevance of the last part of the manuscript:

- Reviewer 1: "The last part of the manuscript deals with the synthesis of even larger polymers but appears somehow out of scope of the study since this is no longer about sequence-defined species but homopolymers made of P or L units and study of some of their macroscopic properties."

- Reviewer 2: "Characterisation of the sequences, as much as the determination of physical parameters of the discrete materials follows literature examples and delivers nothing beyond the already known."

- Reviewer 3: "Showing only the rheological measurements of the LA polymers does also not fit in the flow of the manuscript."

In my opinion, this part is out of the scope of the study described in the manuscript and will remain so until data obtained for copolymers (instead of homopolymers) will be provided, which (as written in the authors' responses to comments" will be part of another study.

Accordingly, I recommend this last part to be removed from the present manuscript.

Additional minor corrections are to be done in all supplementary tables, where "theoretical exact mass" should be changed to "calculated m/z" and all data into parentheses corresponding to "molecular mass" should be removed (they are not relevant here and the reader can question their meaning).

Reviewer #2 (Remarks to the Author):

The authors have given an extensive reply to the issues raised. The technical questions raised by other reviewers have been answered, and additional explanations and some measurements have been performed.

I do, however, still have some issues with regards to novelty. The authors of course stress their advantages, but I still don't think that the work is that different from literature known methods. The answer to novelty is basically a repetition of their main conclusions. While the synthesis of high molecular weight monodisperse with aperiodic sequences is certainly very interesting, it is also possible by other methods. Regardless, the work is of high quality and I do think the work is publishable in Nat Comm.

Reviewer #3 (Remarks to the Author):

I am happy with the revised version and like to recommend publication as it is.

List of corrections and additions

1. Abstract was revised to meet the length requirement.
2. Last sentence in Introduction was revised.
3. Explanations in first paragraph on 6 page were converted in detail and a reference was added (*Polym. Chem.* **2**, 289-299 (2011)).
4. '**Scalability of the convergent synthesis of PAH**' section was fully revised. The contents about the synthesis of monodisperse poly(*rac*-lactic acid) and their physical properties were removed. The section was revised to indicate the synthetic scalability of our method to show the potential to increase the data storage capacity of PAHs.
5. Figure 5 was revised and figure 6 was removed. References and the Supplementary information were revised in accordance to the revised section '**Scalability of the convergent synthesis of PAH**'
6. The Supplementary information concerning 'MALDI-TOF MS/MS sequencing of PcLs' was revised (All "Theoretical exact mass" were revised to "Calculated m/z". All "Molecular mass" and these values were removed).

Concerning the comment from the reviewer 1

The last section of our original manuscript deals with the synthesis of monodisperse poly(α -hydroxy acid)s by the convergent method up to 38kDa molecular weight. The section also contains our physical characterization of these monodisperse poly(lactide)s. Although we believe that the section is informative as the set of characterization is the first example indicating how monodisperse high molecular-weight polymer behave, we also admit that the section might not be coherent to our main conclusion of the manuscript. Therefore, in accordance to the comments from the reviewer 1, we removed the section concerning the physical properties of monodisperse poly(α -hydroxy acid)s from the manuscript. According to this revision, we revised figures (figure 6 is removed and figure5 is revised). References and the Supplementary information were revised accordingly. We also made corrections in the supplementary information as reviewer indicated.